# Transcriptomics and Metabolomics Reveal the Critical Genes of Carotenoid Biosynthesis and Color Formation of Goji (*Lycium barbarum* L.) Fruit Ripening

**DOI:** 10.3390/plants12152791

**Published:** 2023-07-27

**Authors:** Feng Wei, Ru Wan, Zhigang Shi, Wenli Ma, Hao Wang, Yongwei Chen, Jianhua Bo, Yunxiang Li, Wei An, Ken Qin, Youlong Cao

**Affiliations:** 1Wolfberry Engineering Research Institute, Ningxia Academy of Agriculture and Forestry Sciences, Yinchuan 750002, China; wanru2008@163.com (R.W.); lyxyb@163.com (Y.L.); 13037986722@163.com (W.A.); qinken7@163.com (K.Q.); youlongchk@163.com (Y.C.); 2Ningxia State Farm A&F Technology Central, Yinchuan 750002, China; mwl7544@163.com (W.M.); nxnkwh@163.com (H.W.); nxcyw@163.com (Y.C.); 15769680239@139.com (J.B.)

**Keywords:** *Lycium barbarum* L., fruit ripening, fruit color, carotenoid, transcriptomics

## Abstract

Carotenoids in goji (*Lycium barbarum* L.) have excellent health benefits, but the underlying mechanism of carotenoid synthesis and color formation in goji fruit ripening is still unclear. The present study uses transcriptomics and metabolomics to investigate carotenoid biosynthesis and color formation differences in N1 (red fruit) and N1Y (yellow fruit) at three stages of ripening. Twenty-seven carotenoids were identified in N1 and N1Y fruits during the M1, M2, and M3 periods, with the M2 and M3 periods being critical for the difference in carotenoid and color between N1 and N1Y fruit. Weighted gene co-expression network analysis (WGCNA), gene trend analysis, and correlation analysis suggest that *PSY1* and *ZDS16* may be important players in the synthesis of carotenoids during goji fruit ripening. Meanwhile, 63 transcription factors (TFs) were identified related to goji fruit carotenoid biosynthesis. Among them, four TFs (*CMB1-1*, *WRKY22-1*, *WRKY22-3*, and *RAP2-13-like*) may have potential regulatory relationships with *PSY1* and *ZDS16*. This work sheds light on the molecular network of carotenoid synthesis and explains the differences in carotenoid accumulation in different colored goji fruits.

## 1. Introduction

Carotenoids, synthesized in the plastids of photosynthetic and sink organs in plants, are essential molecules for photosynthesis and are responsible for natural colors, such as yellow, orange, and red [1,2]. Carotenoids are also vital for human health and nutrition because of their role as vitamin A precursors and antioxidants [3].

Plant carotenoid synthesis begins with the 5-carbon compound isopentyl diphosphate (IPP) production in plastids via the plastidial 2-C-methyl-D-erythritol 4-phosphate (MEP) pathway. First, IPP isomerase (IPI) and geranylgeranyl diphosphate synthase (GGPS) catalyze the formation of geranylgeranyl diphosphate (GGPP) from IPP in the plastid. Subsequently, phytoene synthase (PSY) catalyzes the formation of 15-*cis*-phytoene from two GGPP molecules. The enzymes phytoene desaturase (PDS), zeta-carotene desaturase (ZDS), zeta-carotene isomerase (Z-ISO), and carotenoid isomerase (CRTISO) further convert 15-*cis*-phytoene to lycopene through a series of desaturations and isomerizations. The subsequent reaction of lycopene is critical for carotenoid diversity in carotenoid metabolism. At this point, the carotenoid biosynthetic pathway branches into two pathways (beta, gamma-carotene, and beta, beta-carotene). In the beta, gamma-carotene pathway, zeta-lycopene cyclase (LCYE), beta-lycopene cyclase (LCYB), cytochrome P450-type monooxygenase 97A (CYP97A), and cytochrome P450-type monooxygenase 97C (CYP97C) catalyze the final production of xanthophyll (lutein) from lycopene. The beta, beta-carotene pathway, on the other hand, requires beta-lycopene cyclase (LCYB), beta-carotene hydroxylase (CHYB/crtR), zeaxanthin epoxidase (ZEP), violaxanthin de-epoxidase (VDE), and neoxanthin synthase (NSY) to catalyze the final production of other carotenoids from lycopene [1,4,5].

Goji (*Lycium barbarum* L.) is a valuable economic tree with high medicinal and nutritional properties. Goji fruit has gained popularity as a “super fruit” due to its high concentration of carotenoids, polysaccharides, and flavonoids [6]. Previous studies have shown eleven free carotenoids and seven carotenoid esters in goji fruit extracts [7]. Carotenoids in goji fruit increase as the fruit ripens, eventually accounting for 0.03% to 0.5% of the dried fruit. The primary carotenoid in goji fruit is zeaxanthin dipalmitate, which accounts for 31–56% of the total carotenoids [8,9]. 

The accumulation of the main carotenoids usually determines the color of the fruit. Red tomatoes are rich in lycopene, which accounts for around 85% of their total carotenoids [10]. Orange tomatoes (*beta* mutant) produce increased amounts of beta-carotene at the expense of lycopene [11]. Light-yellow tomatoes, on the other hand, contain little beta-carotene and almost no lycopene [12].

Capsanthin is the most abundant carotenoid in red peppers [13]. However, the yellow pepper varieties contain mainly lutein, alfa-carotene, and beta-carotene, but no capsanthin. Orange peppers accumulate different types of carotenoids, such as capsanthin, lutein, alfa-carotene, and beta-carotene [14]. Beta-carotene levels in carrots and sweet potatoes are typically high [15]. The carotenoid composition of winter squashes varies, with b-carotene, lutein, and violaxanthin being the most abundant [16]. The carotenoid content also varies among goji germplasms. Carotenoids (mainly zeaxanthin) are abundant in red fruits (*Lycium barbarum* L.) but undetectable in black fruits (*Lycium ruthenicum* L.) [17].

Although all genes involved in carotenoid biosynthesis have been isolated and studied in many plants, research on the genes regulating carotenoid biosynthesis in goji fruits is still lacking. Furthermore, the reasons why different goji cultivars differ in color are unknown. In the present study, transcriptomics and metabolomics were used to detect carotenoid changes and gene transcriptome profiles in two goji verities (N1 and N1Y) during fruit ripening. In addition, weighted gene co-expression network analysis (WGCNA), gene trend, and correlation analysis could identify the essential genes and transcription factors involved in carotenoid synthesis. This study provides important insights into the molecular network of carotenoid synthesis of goji fruits and explains the differences in carotenoid accumulation in different colored goji fruits.

## 2. Results

### 2.1. Carotenoid Differences in N1 and N1Y during Fruit Ripening

A previous study identified a bud mutant cultivar N1Y of N1, which has similar growth characteristics and fruit size to N1. However, the fruit color of N1Y is yellow, while N1 is red (Figure 1a). This could be due to a difference in carotenoid levels between N1 and N1Y during fruit ripening. We analyzed fresh fruit samples from the two cultivars at different stages of fruit ripening, namely M1, M2, and M3 (Figure 1a), to compare the carotenoid profile in N1 and N1Y. Total carotenoid content in N1 and N1Y continued to increase during fruit ripening, and N1Y fruit had significantly higher carotenoid levels than N1 during the M2 and M3 periods (Figure 1b). Based on the principal component analysis (PCA) of carotenoids, sampling of N1 and N1Y at different stages was readily repeatable and could be used for subsequent analysis (Figure 1c).

Our study identified 27 carotenoids in the fruits of N1 and N1Y (Figure 2 and Appendix A and Appendix A). Phytoene, zeaxanthin, zeaxanthin dipalmitate, zeaxanthin palmitate, beta-cryptoflavin palmitate, and rubixanthin palmitate were among those whose levels rose during fruit ripening. In contrast, beta-carotene and xanthophyll levels decrease dramatically during fruit ripening (Figure 2). During the M1 period, xanthophyll was the major carotenoid in N1 and N1Y fruits, accounting for 57.2% and 59.7% of the total carotenoid content, respectively. According to the PCA analysis for the M1 period, PC1 (95.27%) has a well explained variance, and xanthophyll was the main loading component (Appendix A). However, there was no significant difference in xanthophyll levels between N1 and N1Y (Figure 2). During the M2 period, zeaxanthin and zeaxanthin dipalmitate were the main carotenoids in N1 and N1Y fruits, accounting for 56.2% and 77.3% of the total carotenoid content, respectively. PCA analysis for M2 showed that PC1 (99.81%) provided a well explained variance, and zeaxanthin was the major loading component (Appendix A). N1Y fruits contain more zeaxanthin than N1, while N1 has more zeaxanthin dipalmitate (Figure 2). Phytoene, zeaxanthin palmitate, and zeaxanthin dipalmitate were the main carotenoids of N1 fruits during the M3 period, accounting for 59.6% of the total carotenoid content. Zeaxanthin, phytoene, and zeaxanthin palmitate were identified as the main carotenoids of N1Y fruit, accounting for 87.9% of the carotenoid content. According to the PCA analysis for the M3 period, PC1 (98.71%) has a well explained variance, and zeaxanthin and phytoene were the main loading components (Appendix A). N1Y had significantly higher levels of zeaxanthin and phytoene than N1. However, N1 had higher levels of zeaxanthin palmitate and zeaxanthin dipalmitate than N1Y (Figure 2). 

Thus, the M2 and M3 periods were critical for the carotenoid content difference between N1 and N1Y fruit. Furthermore, differences in the main carotenoids can explain the color variation between N1 and N1Y fruit at the M2 and M3 periods. Much of the zeaxanthin in the N1 fruit was converted to zeaxanthin palmitate and zeaxanthin dipalmitate during the M2 and M3 periods, while the N1Y fruit continued to accumulate zeaxanthin. As a result, from M2 to M3, the N1 fruit gradually turned red, whereas the N1Y fruit turned yellow.

### 2.2. Transcriptome Profiles of N1 and N1Y Fruits in Different Ripening Periods

The gene expression profiles of N1 and N1Y fruits at the M1, M2, and M3 periods were examined using RNA sequencing to identify the essential genes for carotenoid biosynthesis at various ripening stages. We obtained an average of 7 G clean bases per sample, with 93.38% of bases scoring Q30 (Appendix A). Over 92% of the reads mapped to the reference genome of goji (Appendix A). Based on principal component analysis (PCA), the samples of N1 and N1Y were highly repeatable at different periods and could be used for subsequent analysis. The cumulative contribution of PC1 reached 87.27%, which may explain the gene expression difference in M1, M2, and M3 of N1 and N1Y fruits (Figure 3a). Finally, 35,615 genes were expressed in N1 and N1Y fruits. There were 1000 differentially expressed genes (DEGs) between N1-M1 and N1Y-M1, 116 DEGs between N1-M2 and N1Y-M2, and 314 DEGs between N1-M3 and N1Y-M3 (Figure 3b, Appendix A). There were 9666 DEGs in N1-M1 vs. N1-M2, 9835 DEGs in N1-M1 vs. N1-M3, and 2712 DEGs in N1-M2 vs. N1-M3 (Figure 3c, Appendix A). N1-M1 vs. N1-M2, N1-M vs. N1-M3, and N1-M2 vs. N1-M3 all share 883 common DEGs, which were enriched in proteins involved in glyoxylate and dicarboxylate metabolism, carbohydrate metabolism, phenylpropanoid biosynthesis, carotenoid biosynthesis, cysteine and methionine metabolism, and photosynthesis proteins (Figure 4a). The DEGs for N1Y-M1 vs. N1Y-M2, N1Y-M1 vs. N1Y-M3, and N1Y-M2 vs. N1Y-M3 were 9549, 10,701, and 3501, respectively (Figure 3d, Appendix A). N1Y-M1 vs. N1Y-M2, N1Y-M1 vs. N1Y-M3, and N1Y-M2 vs. N1Y-M3 shared 943 common DEGs, which were primarily involved in alpha-linolenic acid metabolism, photosynthesis, carotenoid biosynthesis, and starch and sucrose metabolism (Figure 4b). The difference between N1 and N1Y at the M2 and M3 periods was significantly responsible for the change in carotenoid content and the resulting fruit color. Therefore, the DEGs of N1-M1 vs. N1-M2, N1-M1 vs. N1-M3, N1Y-M1 vs. N1Y-M2, and N1Y-M1 vs. N1Y-M3 were analyzed. Finally, 5614 DEGs were found in signaling and cellular processes, carotenoid biosynthesis, and carbohydrate, starch, sucrose, and amino acid metabolism (Figure 4c). We selected 13 genes (Appendix A) to validate our RNA-seq data. qRT-PCR was used to examine the expression levels of these genes in N1 and N1Y during M1, M2, and M3 (Appendix A). The results revealed that the RNA data were valid and trustworthy, as the expression patterns of these genes matched the RNA-seq results.

### 2.3. Expression Profiles of Genes of the Carotenoid Biosynthetic Pathway during Goji Fruit Ripening

To further investigate the critical genes of carotenoid biosynthesis during goji fruit ripening, we analyzed the expression of the carotenoid biosynthesis genes in N1 and N1Y fruits (Figure 5). Three phytoene synthase genes (*PSY1*, *PSY2-1,* and *PSY2-2*) involved in the condensation of geranylgeranyl diphosphate (GGPP) were expressed during goji fruit ripening (Figure 5, Appendix A). Among them, *PSY1* showed a significant increase during goji fruit ripening (Figure 6a), with its expression tightly linked to total carotenoid, zeaxanthin dipalmitate, beta-carotene, and xanthophyll (Figure 6b, Appendix A). Expression of *PSY2-1* and *PSY2-2* significantly decreased during goji fruit ripening (Figure 6a) and were highly correlated with zeaxanthin dipalmitate, beta-carotene, and xanthophyll (Figure 6b, Appendix A). Three phytoene dehydrogenases genes (*PDS1*, *PDS2*, and *PDS3*) were discovered in the transcriptome profiles of N1 and N1Y fruit. Only *PDS1* and *PDS3* were expressed during the catalytic conversion of phytoene to 15,9′-tri-cis-phytofluene and 9,15,9′-tri-cis-zeta-carotene (Figure 5, Appendix A). *PDS1* levels rose throughout goji fruit ripening (Figure 6a) and significantly correlated with zeaxanthin dipalmitate, xanthophyll, total carotenoid, and beta-carotene (Figure 6b, Appendix A). A 15-cis-zeta-carotene isomerase gene (*Z-ISO*) catalyzing the conversion of 9,15,9′-tri-cis-zeta-carotene to 9,9′-di-cis-zeta-carotene was also identified and showed upregulation (Figure 5 and Figure 6a and Appendix A). Meanwhile, the expression of *Z-ISO* was strongly correlated with xanthophyll, beta-carotene, and zeaxanthin dipalmitate content (Figure 6b, Appendix A). Sixteen zeta-carotene desaturases genes (*ZDSs*) catalyzing the conversion of 9,9′-di-cis-zeta-carotene to 7,9,9′-tri-cis-neurosporene and 7,9,7,9′-tetra-cis-lycopene were also detected (Figure 5, Appendix A). *ZDS12* and *ZDS16* were significantly upregulated (Figure 6a), and their expression was strongly correlated with total carotenoid, zeaxanthin, phytoene, xanthophyll, beta-carotene, and zeaxanthin dipalmitate levels (Figure 6b, Appendix A). Eight *ZDSs* (*ZDS1*, *ZDS2*, *ZDS5*, *ZDS6*, *ZDS7*, *ZDS10*, *ZDS 11*, and *ZDS13*) were significantly downregulated (Figure 6a) and inversely associated with total carotenoid concentration (Figure 6b, Appendix A). Four prolycopene isomerases genes (*crtISOs*) were identified and catalyzed the enzymatic conversion of 7,9,7,9′-tetra-cis-lycopene to lycopene (Figure 5, Appendix A). Only *crtISO-4* varied significantly throughout fruit ripening and strongly correlated with zeaxanthin and beta-carotene (Figure 6a,b, Appendix A). One lycopene epsilon cyclase gene (*LCYE*) catalyzing lycopene to delta-carotene was also detected (Figure 5, Appendix A) and was downregulated during fruit ripening (Figure 6a). The expression of *LCYE* was strongly correlated with total carotenoid, beta-carotene, xanthophyll, and zeaxanthin dipalmitate (Figure 6b, Appendix A). We also detected one lycopene beta-cyclase gene (*LCYB*) in the transcriptome profiles of N1 and N1Y fruit (Figure 5, Appendix A). The expression of *LCYB* increased during fruit ripening (Figure 6a) and was strongly correlated with xanthophyll, beta-carotene, and zeaxanthin dipalmitate levels (Figure 6b, Appendix A). The transcriptome profiles of N1 and N1Y fruit revealed three cytochrome P450-type monooxygenase 97A genes (*CYP97As*) (Figure 5, Appendix A). *CYP97A2* and *CYP97A3* levels increased significantly during fruit ripening (Figure 6a) and were closely linked with zeaxanthin, xanthophyll, beta-carotene, and zeaxanthin dipalmitate (Figure 6b, Appendix A). Two beta-carotene hydroxylases genes (*CrtRs*) were expressed during goji fruit ripening (Figure 5, Appendix A). *CrtR1* significantly decreased during goji fruit ripening, while *CrtR2* increased (Figure 6a). Expression levels of *CrtR1* and *CrtR2* strongly correlated with total carotenoid, zeaxanthin, xanthophyll, beta-carotene, and zeaxanthin dipalmitate (Figure 6b, Appendix A). Three cytochrome P450-type monooxygenase 97C genes (*CYP97Cs*) were involved in the transcriptome profiles (Figure 5, Appendix A), with no significant change during goji fruit ripening (Figure 6a). Three zeaxanthin epoxidase genes (*ZEPs*) were also expressed during goji fruit ripening (Figure 5, Appendix A). All of them decreased during fruit ripening (Figure 6b), and expression of *ZEP3* was strongly correlated with xanthophyll, beta-carotene, and zeaxanthin dipalmitate content (Figure 6b, Appendix A). The transcriptome profiles identified two violaxanthin de-epoxidases genes (*VDEs*) (Figure 5, Appendix A). *VDE2* decreased during goji fruit ripening (Figure 6a), and the expression levels correlated with xanthophyll, beta-carotene, and zeaxanthin dipalmitate (Figure 6b, Appendix A). One neoxanthin synthase gene (*NSY*) was significantly elevated during goji fruit ripening (Figure 5, Figure 6a, Appendix A), and its expression was linked with zeaxanthin, xanthophyll, beta-carotene, and zeaxanthin dipalmitate (Figure 6b, Appendix A).

According to the expression and correlation, we think that *PSY1, PDS1, Z-ISO, ZDS16, crtISO4, LCYE, LCYB, CYP97A2, CYP97A3, CrtR2, VDE2, ZEP3,* and *NSY* may play important roles in carotenoid biosynthesis during goji fruit ripening. Moreover, the expression of *PSY1* (0.80, *p* < 0.05) and *ZDS16* (0.88, *p* < 0.05) had a strong positive correlation with the total carotenoid content of N1 and N1Y fruit (Figure 6b, Appendix A). Thus, we conclude that *PSY1* and *ZDS16* were more important for goji fruit carotenoid biosynthesis than *PDS1, Z-ISO, crtISO4, LCYE, LCYB, CYP97A2, CYP97A3, CrtR2, VDE2, ZEP3*, and *NSY*.

### 2.4. Identification of WGCNA Modules of Carotenoid Biosynthesis in Goji Fruit

A weighted gene co-expression network analysis (WGCNA) was performed for all N1 and N1Y fruit gene expressions to identify critical genes involved in carotenoid biosynthesis, and 17 WGCNA modules were identified (Figure 7a, Appendix A). We discovered that the main carotenoids of N1 and N1Y fruits at the M1, M2, and M3 stages were zeaxanthin, phytoene, xanthophyll, zeaxanthin dipalmitate, zeaxanthin palmitate, beta-cryptoflavin palmitate, and rubixanthin palmitate. Therefore, we primarily sought modules highly related to these substances. According to the WGCNA, the yellow (0.94, *p* < 0.05), brown (0.70, *p* < 0.05), and tan modules (0.65, *p* < 0.05) had a positive correlation with total carotenoid content, while the pink (−0.9, *p* < 0.05) and light cyan modules (−0.64, *p* < 0.05) had a negative correlation. The cyan (0.73, *p* < 0.05), brown (0.69, *p* < 0.05), and green modules (0.64, *p* < 0.05) had a positive correlation with zeaxanthin content, while the blue (−0.61, *p* < 0.05) and turquoise modules (−0.61, *p* < 0.05) had a negative correlation. The yellow (0.9, *p* < 0.05) and tan modules (0.84, *p* < 0.05) were correlated positively with phytoene content, whereas the pink (−0.83, *p* < 0.05) and red modules (−0.81, *p* < 0.05) were correlated negatively. The turquoise (0.78 and 0.84, *p* < 0.05) and blue modules (0.82 and 0.84, *p* < 0.05) had a positive correlation with xanthophyll and beta-carotene content, while the brown (−0.99 and −0.95, *p* < 0.05) and yellow modules (−0.73 and −0.6, *p* < 0.05) had a negative correlation. The brown module (0.96, *p* < 0.05) had a positive correlation with zeaxanthin dipalmitate content, while blue and turquoise modules (−0.81, *p* < 0.05) had a negative correlation. The purple module (0.93, *p* < 0.05) also had a positive correlation with zeaxanthin palmitate content, while the pink module (−0.6, *p* < 0.05) had a negative correlation. The content of beta-cryptoflavin palmitate and rubixanthin palmitate was correlated positively with the purple module (0.94, *p* < 0.05) and negatively with the pink module (−0.67, *p* < 0.05).

*PSY1* and *ZDS16* were found to be essential genes for goji fruit carotenoid biosynthesis during fruit ripening. The brown module consisted of *PSY1* and *ZDS16* (Appendix A). Moreover, total carotenoid, zeaxanthin, xanthophyll, beta-carotene, zeaxanthin dipalmitate, neoxanthin, violaxanthin, and lutein palmitate was also strongly correlated with the brown module (Figure 7a). Therefore, the genes of the brown module were worthy of further analysis and study. Meanwhile, we also used the Mfuzz software to examine the expression trend of genes in goji fruit at various stages of ripening. The findings revealed that differential genes could be divided into six clusters (Figure 7b, Appendix A). Among them, *PSY1* and *ZDS16* were classified in cluster 2.

### 2.5. Identification of Key Transcription Factors Associated with Carotenoid Biosynthesis Pathway

Many studies have shown that transcription factors (TFs) play vital regulatory functions in the biosynthesis of carotenoids in plants [1,4]. The genes of the brown module were worthy of further analysis and study. As a result, we conclude that some TFs in the brown module may be involved in the regulation of carotenoid synthesis. We identified 63 TFs from the brown module (Appendix A) that can be classified into two groups based on their expression levels (Figure 8). Since *PSY1* and *ZDS16* were more important for carotenoid biosynthesis in goji fruit during ripening, we propose that TFs meet the following three criteria to be considered as potential regulators of carotenoid synthesis. First, the correlation with *PSY1* and *ZDS16* must be more than 0.9 or less than −0.9 (Appendix A). There should also be more than a 2-fold increase or decrease in the expression in M2 or M3 compared to M1 (Figure 8). Lastly, the network weight values with *PSY1* and *ZDS16* must be greater than 0.5 (Appendix A). We finally selected three TFs (*CMB1-1*, *WRKY22-1*, and *WRKY22-3*) that positively regulate the carotenoid synthesis and one TF (*RAP2-13-like*) that negatively regulates the carotenoid synthesis (Figure 9).

## 3. Discussion

As a traditional Chinese medicinal and nutritional plant, goji (*Lycium barbarum* L.) has attracted much interest from scientists worldwide, especially regarding the extraction of goji carotenoids and their role in human health [18]. Previous studies discovered 18 carotenoids in goji fruit extracts [7]. Using liquid chromatography–tandem mass spectrometry (LC-MS/MS), the current study identified 27 carotenoids in N1 and N1Y fruits during fruit ripening (Figure 2 and Appendix A). Peng, Ma, Li, Leung, Jiang and Zhao [9] indicated that the carotenoid content in goji berries increases as the fruit ripens, and zeaxanthin dipalmitate, which makes up 31–56% of all the carotenoids in fruits, is the primary carotenoid. In the present study, we further analyzed the carotenoids content in N1 and N1Y in the three stages of fruit ripening from M1 to M3 (Figure 1b). In N1 and N1Y fruits in the M1 period, xanthophyll was the main carotenoid, accounting for 57.2% and 59.7% of the total carotenoid content, respectively (Figure 2). Zeaxanthin and zeaxanthin dipalmitate, which together accounted for 56.2% and 77.3% of the carotenoid content, respectively, were the primary carotenoids in N1 and N1Y fruits during the M2 period (Figure 2). At the M3 period, the main carotenoid components of N1 fruits were phytoene, zeaxanthin palmitate, and zeaxanthin dipalmitate, which made up 59.6% of the total carotenoid content. However, the primary carotenoids in N1Y fruit were zeaxanthin, phytoene, and zeaxanthin palmitate, which accounted for 87.9% of the carotenoid content (Figure 2). Liu, Zeng, Sun, Wu, Hu, Shen and Wang [17] compared the carotenoid accumulation in red fruit (*Lycium barbarum* L.) and black fruit (*Lycium* ruthenicum Murr.). They found that the principal carotenoids in red and black fruit at the green fruit stage (S1) are xanthophyll, violaxanthin, and β-carotene, but, as the fruit ripens, these levels decrease. In addition, the zeaxanthin content of red fruit significantly increases from color break (S2) to the ripe fruit stage (S4) but is undetectable in the black fruit. The present study further explores the carotenoid changes in goji fruit during ripening. As the N1 and N1Y fruits ripened, we observed an increase in phytoene, zeaxanthin, zeaxanthin dipalmitate, zeaxanthin palmitate, beta-cryptoflavin palmitate, and rubixanthin palmitate. Moreover, the amounts of beta-carotene and xanthophyll in N1 and N1Y fruit considerably reduced from M1 to M3 (Figure 2).

Fruit color, an essential appearance and quality character, is determined by the buildup of different pigment compounds. The most significant among them is carotenoid accumulation, which allows plants to take on yellow, orange, and red colors [10,13,19,20,21,22]. Most goji cultivars have red fruit color, while a few are black, yellow, purple, and white (Appendix A). So far, the cause of the color variation between different goji cultivars is still unknown. Some studies suggest that the color difference between red and black fruits is due to the accumulation of anthocyanins in black fruits and carotenoids in red fruits [2,23]. In the present study, the M2 and M3 periods were critical for the difference in carotenoid content and fruit color between N1 and N1Y fruits (Figure 1a,b). During the M2 and M3 periods, zeaxanthin in the N1 fruits was largely converted to zeaxanthin palmitate and zeaxanthin dipalmitate, while the N1Y fruits continued to accumulate zeaxanthin. Thus, the color of the N1 fruit gradually turned red, while the N1Y fruit color appeared yellow from the M2 to M3 period (Figure 1c).

The genes involved in carotenoid metabolism have been identified and investigated in several plants [2,24]. The genes involved in plant carotenoid biosynthesis usually have multiple copies that exhibit tissue-specific expression, such as *PSY1*, *PSY2*, and *PSY3*, which are specifically expressed in the fruits, leaves, and roots of tomatoes and citrus, respectively [25,26]. All tissues express *PSY-A*, whereas *PSY-B* is found only in the leaves and roots of watermelons [19]. Most citrus species have two *PDS* genes and more than three *ZDS* genes [27]. At least two *LCYBs* and two *LCYEs* are present in tomatoes, papaya, and citrus. The *LCYB1* gene of tomato is highly expressed in vegetative tissues, while the *LCYB2* gene is significantly expressed in fruits and flowers [28,29,30,31]. However, few reports about goji carotenoid metabolism exist. Zhao, et al. [32] cloned the *LcPSY*, *LcPDS*, *LcZDS*, *LcLCYB*, *LcLCYE*, *LcCHXE*, *LcZEP*, and *LcNCED* from *Lycium* chinense, as those genes were constitutively expressed in the leaves, flowers, and fruits. Liu, Zeng, Sun, Wu, Hu, Shen and Wang [17] found that similar enzyme activities of *PSY1*, *CYC-B*, and *CRTR-B2* were observed in red fruits (*Lycium barbarum* L.) and black fruits (*Lycium ruthenicum* Murr.), suggesting that the undetectable carotenoid content in black fruits was not due to inactivation of carotenoid biosynthetic enzymes. In addition, expression of carotenoid cleavage dioxygenase 4(*CCD4*) was higher in black fruit than in red fruit.

In the current study, we used RNA sequencing to evaluate the gene expression profiles of N1 and N1Y fruits at M1, M2, and M3 periods (Figure 3a). A total of 35,615 genes were expressed in N1 and N1Y fruits during fruit ripening. Based on the gene expression levels, we conclude that *PSY1*, *PDS1*, *Z-ISO*, *ZDS16, crtISO4, LCYE, LCYB, CYP97A2, CYP97A3, CrtR2, VDE2, ZEP3,* and *NSY* may play vital roles in the synthesis of carotenoids during goji fruit ripening (Figure 6a, Appendix A). Furthermore, *PSY1* and *ZDS16* were correlated more strongly with the total carotenoid content of goji fruit than *PDS1, Z-ISO, crtISO4, LCYE, LCYB, CYP97A2, CYP97A3, CrtR2, VDE2, ZEP3,* and *NSY* (Figure 6b, Appendix A). However, during fruit ripening, there was no significant difference in the expression of *PSY1* and *ZDS16* between N1 and N1Y (Figure 6a). Thus, variations in the expression levels of *PSY1* and *ZDS16* may not be the cause of the variations in carotenoids and color between N1 and N1Y fruits. Since it is unclear how zeaxanthin converts to zeaxanthin palmitate and zeaxanthin dipalmitate in goji fruit, the mechanism needs further investigation.

The regulation of plant carotenoids is challenging due to changes in concentration and composition during development. Numerous studies have demonstrated that TFs regulate carotenoid accumulation by influencing plant growth [2]. In tomato fruit, the MADS-box transcription factor *RIN* influences carotenoid accumulation by directly upregulating *PSY1, ZISO, ZDS,* and *CRTISO* [33,34]. Zhang, et al. [35] reported that *SICMB1* regulates ethylene production and carotenoid accumulation during tomato fruit ripening via interactions with *SIMADS-RIN, SIMADS1, SIAP2a,* and *TAGL1*. In dark-grown *Arabidopsis*, phytochrome-interacting factor 1 (*PIF1*) can bind the *PSY* promoter and inhibit carotenoid accumulation [36]. Similarly, ethylene-responsive transcription factor *AtPAP2.2* can bind to the promoters of *PDS* and *PYS* and influence carotenoid biosynthesis [11]. Yuan, et al. [37] indicated that *SIWRKY35* could positively regulate the synthesis of tomato carotenoids by directly inducing *SIDXS1* to rewire metabolism toward the MEP pathway. Hu, et al. [38] hypothesized that *bHLH, MYB, NAC, MADS,* and *WRKY* are involved in the biosynthesis of carotenoids and anthocyanins in Coffea arabica. There are few studies on the TFs responsible for carotenoid production in goji fruits. Yin, et al. [39] reported that two *R2R3-MYBs* and two *BBXs* may play key roles in the carotenoid biosynthesis of goji. In the present study, we analyzed the gene expression of N1 and N1Y fruit during fruit ripening by WGCNA, and seventeen WGCNA modules were identified (Figure 7a, Appendix A). Among them, brown modules include *PSY1* and ZDS16 (Appendix A), and strongly correlate with total carotenoid, zeaxanthin, phytoene, xanthophyll, zeaxanthin dipalmitate, zeaxanthin palmitate, beta-cryptoflavin palmitate, and rubixanthin palmitate (Figure 7a). Thus, we conclude that many TFs in the brown module regulate carotenoid synthesis in goji fruit. We identified 63 TFs from the brown module (Appendix A) that may be divided into 2 groups based on expression levels (Figure 8). According to the TF expression, network weight values, and correlation with *PSY1* and ZDS16, we conclude that one MADS-box TF (CMB1-1) and two WRKY TFs (WRKY22-1 and WRKY22-3) may favorably influence the synthesis of carotenoids by regulating *PSY1* and ZDS16. One ethylene-responsive TF (RAP2-13-like) may negatively control the synthesis of carotenoids by *PSY1* and ZDS16 (Figure 9).

## 4. Materials and Methods

### 4.1. Plant Materials

Ningqi 1 (N1) and the bud mutant of Ningqi 1 (N1Y) were from Center of Wolfberry Engineering Technology Research, Yinchuan, Ningxia, China. Fruits of N1 and N1Y were collected at 15 (M1, green fruit), 25 (M2, color break), and 40 (M3, ripe fruit) days after flowering. All samples were immediately ground in liquid nitrogen and stored at −80 °C until needed.

### 4.2. Determination of Carotenoid Content

The samples were prepared and extracted by Metware Biotechnology Co., Ltd. “http://www.metware.cn/ (accessed on 5 May 2022)” and analyzed using the ultra-high-performance liquid chromatography–high-resolution mass spectrometry (UHPLC-HRMS) system. Chromatographic separation of the carotenoid extract was performed on a C30 column (2.0 mm × 100 mm, 3.0 µm, YMC Co., Ltd., Shanghai, China) and analyzed by ExionLC™ AD UHPLC (AB SCIEX, Framingham, MA, USA) at 28 °C. At a flow rate of 0.8 mL/min, a binary mobile phase consisting of eluent A: methanol/acetonitrile (1:3, *v*/*v*) with 0.01% butylated hydroxytoluene (BHT) and 0.1% formic acid, and eluent B: methyl tert-butyl ether with 0.01% BHT, was used. The gradient was as follows: 0% B for 0–3 min; 0–42% B for 3–6 min; 42–80% B for 6–8 min; 80–95% B for 8–9 min. The column was then washed and re-equilibrated with 0% B for 2 min. The injection volume for each sample was 2 µL. Mass spectrometry ionization was performed in positive mode on a high-resolution tandem mass spectrometer-AB QTRAP 6500 System (AB SCIEX, Framingham, MA, USA) equipped with an atmospheric pressure chemical ionization (APCI) turbo-ion spray interface. The parameters for APCI source operation were the same as in the previous study [5]. MRM analysis was performed by Metware Biotechnology Co., Ltd. (Wuhan, China). Calibration curves for 27 standards were used to quantify carotenoids. Carotenoid standard solutions of 0.01 μg/mL, 0.05 μg/mL, 0.1 μg/mL, 0.5 μg/mL, 1 μg/mL, 5 μg/mL, 10 μg/mL, and 40 μg/mL were prepared, and the mass spectral peak intensity data of the corresponding quantitative signals of the standards at each concentration were obtained (Appendix A). Data analysis was performed using Analyst 1.6.3 software (AB SCIEX).

### 4.3. Transcriptomic Analysis

The transcriptomic analysis method used in this study was modified from a previous study [23]. In brief, the clean reads were mapped to the genome of goji [40] using HISAT2 [41]. Differential expression was analyzed using DESeq 2 [42], and genes that met the criteria |log2 fold change| ≥ 1 and false discovery rate (FDR) < 0.05 were classified as differentially expressed. The raw RNA-seq data are freely available from the National Center for Biotechnology Information (accession: PRJNA946044).

### 4.4. Quantitative Real-Time PCR (qRT-PCR) Analysis

RNA extraction and qRT-PCR were performed as previously described [39]. Primers were designed using NCBI-Primer-Blast “https://www.ncbi.nlm.nih.gov/tools/primer-blast/ (accessed on 5 June 2022)” (Appendix A). For each sample, three replicates were performed. Quantitative data were analyzed using the 2^−ΔCT^ method.

### 4.5. Weighted Gene Co-Expression Network Analysis (WGCNA)

R package WGCNA [43] was used for identifying highly co-expressed gene modules with carotenoid content. WGCNA network construction and module detection were performed using an unsigned topological overlap matrix (TOM), a power β of 7, a deep split of 2, a minimum module size of 30, and a branch merge cut height of 0.25. The module eigengene value was calculated and used to evaluate the association of modules with carotenoid content in the eighteen samples.

### 4.6. Gene Expression Trend Analysis

R package Muffz (10.18129/B9.bioc.Mfuzz) was used to analyze gene expression trends in all samples. The c parameter was set to 6, and M-estimate was used for m.

### 4.7. Statistical Analysis

Statistical analysis was performed using Microsoft Office Excel 2013 and SPSS 20.0 (IBM Corporation, Armonk, NY, USA), GraphPad Prism 9.0 (GraphPad Software, Inc., 7825 Fay Avenue, Suite 230, La Jolla, CA, USA), and R (http://www.r-project.org/) [44].

## 5. Conclusions

In the present study, the difference in carotenoid biosynthesis and color formation in N1 (red fruit) and N1Y (yellow fruit) during fruit ripening were revealed by transcriptomics and carotenoid profiling. Finally, 27 carotenoids were identified in N1 and N1Y fruits at the M1, M2, and M3 periods. Among them, M2 and M3 were the critical periods for carotenoid and color differences between N1 and N1Y fruit. The zeaxanthin in the N1 fruits was significantly converted to zeaxanthin palmitate and zeaxanthin dipalmitate from M2 to M3, while the N1Y fruits continued accumulating zeaxanthin. As a result, N1 fruits progressively take on a reddish hue, while the color of N1Y fruits appears yellow from M2 to M3. However, the mechanism of conversion of zeaxanthin into zeaxanthin palmitate and zeaxanthin dipalmitate in goji fruits necessitates further investigation. Our findings from WGCNA analysis, gene trend analysis, and correlation analysis suggest that *PSY1* and *ZDS1* may be crucial genes in the synthesis of carotenoids during goji fruit ripening. Additionally, there are four TFs (*CMB1-1, WRKY22-1, WRKY22-3,* and *RAP2-13-like*) that may be involved in the carotenoid biosynthesis of goji fruit by regulating *PSY1* and *ZDS1*. This work provides a new understanding of the molecular network of carotenoid synthesis and explains the variations in carotenoid accumulation in different colored goji fruits.

## Figures and Tables

**Figure 1 plants-12-02791-f001:**
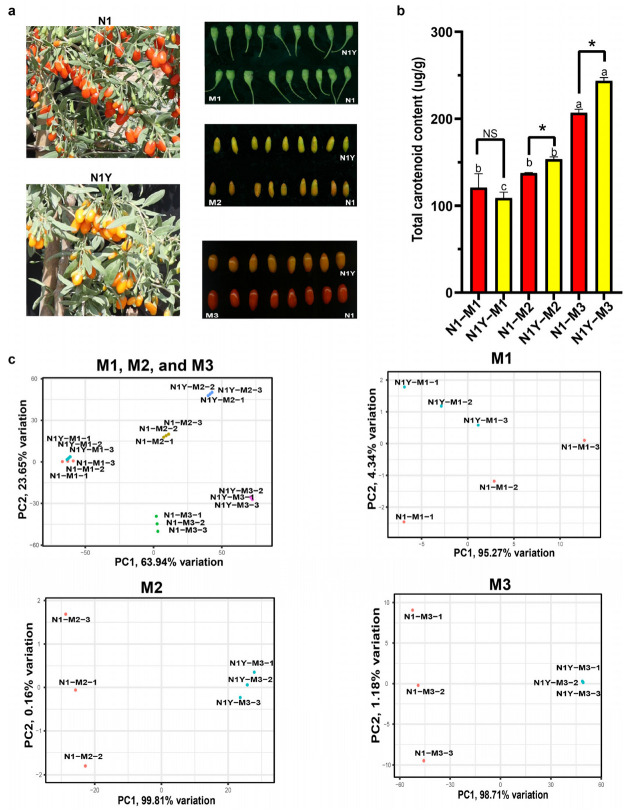
The change in carotenoid content in the fruits of N1 and N1Y in the different ripening periods. (**a**) The phenotype of N1 and N1Y fruits. (**b**) Comparison of total carotenoid content of N1 and N1Y fruits in different periods. Small letters indicate that N1 or N1Y have significant differences between M1, M2, and M3 (*p* < 0.05), as analyzed by Duncan’s multiple tests; * means a significant difference between N1 and N1Y at M1, M2, and M3 periods (*p* < 0.05), as analyzed by Duncan’s multiple tests; NS means no significant difference. (**c**) Principal component analysis for carotenoid content of N1 and N1Y fruits at the M1, M2, and M3 periods.

**Figure 2 plants-12-02791-f002:**
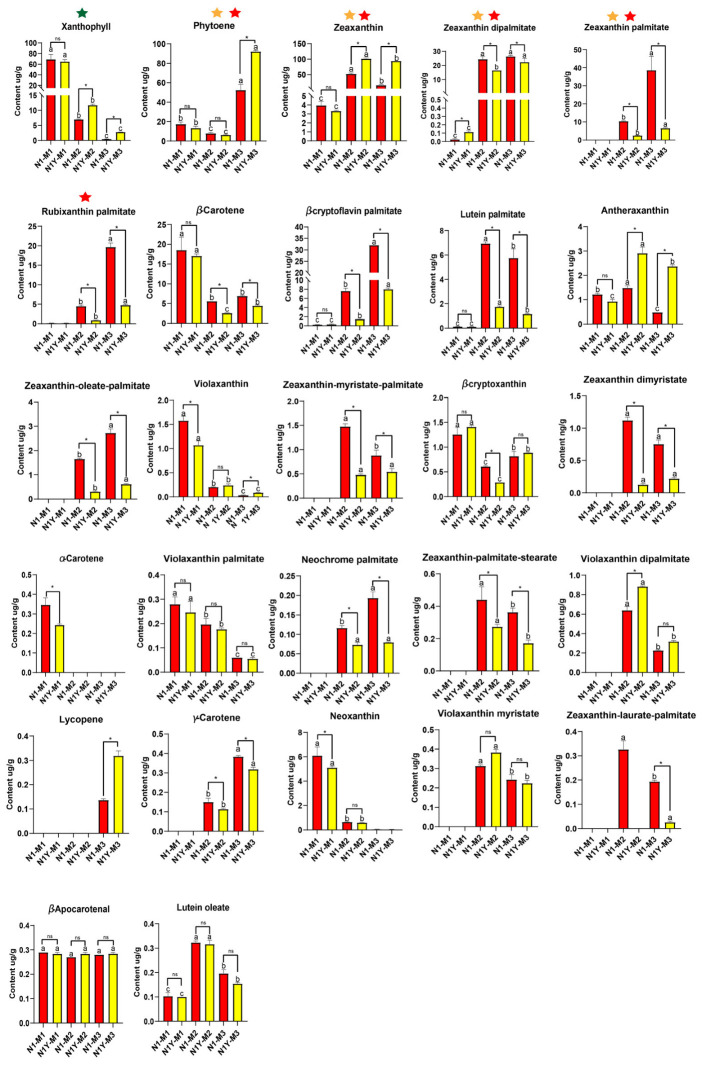
The change in individual carotenoid content in N1 and N1Y fruits at different ripening stages. Small letters indicate that N1 or N1Y have significant differences between M1, M2, and M3 (*p* < 0.05), as analyzed by Duncan’s multiple tests; * means a significant difference between N1 and N1Y at M1, M2, and M3 periods (*p* < 0.05), as analyzed by Duncan’s multiple tests; NS means no significant difference. The green, orange, and red asterisks represent the primary carotenoid of N1 and N1Y fruits at the M1, M2, and M3 periods, respectively.

**Figure 3 plants-12-02791-f003:**
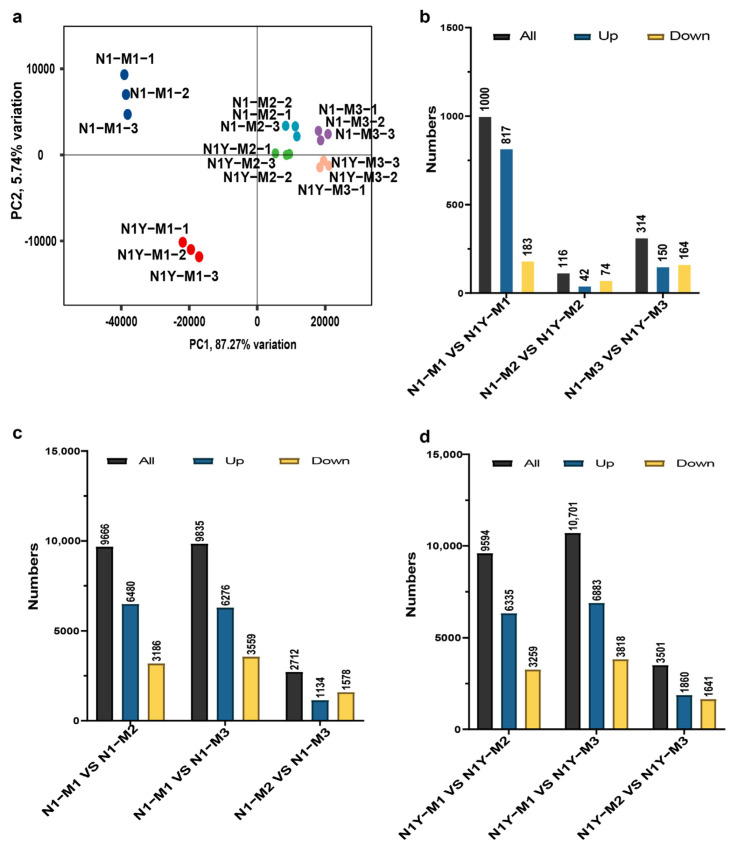
Transcriptome profiles of N1 and N1Y fruit at different stages of ripening. (**a**) The principal component analysis for the gene expression of N1 and N1Y fruits at M1, M2, and M3. (**b**) The number of differentially expressed genes (DEGs) between N1 and N1Y at M1, M2, and M3. (**c**) The number of differentially expressed genes (DEGs) of N1 between M1, M2, and M3. (**d**) The number of differentially expressed genes (DEGs) of N1Y between M1, M2, and M3.

**Figure 4 plants-12-02791-f004:**
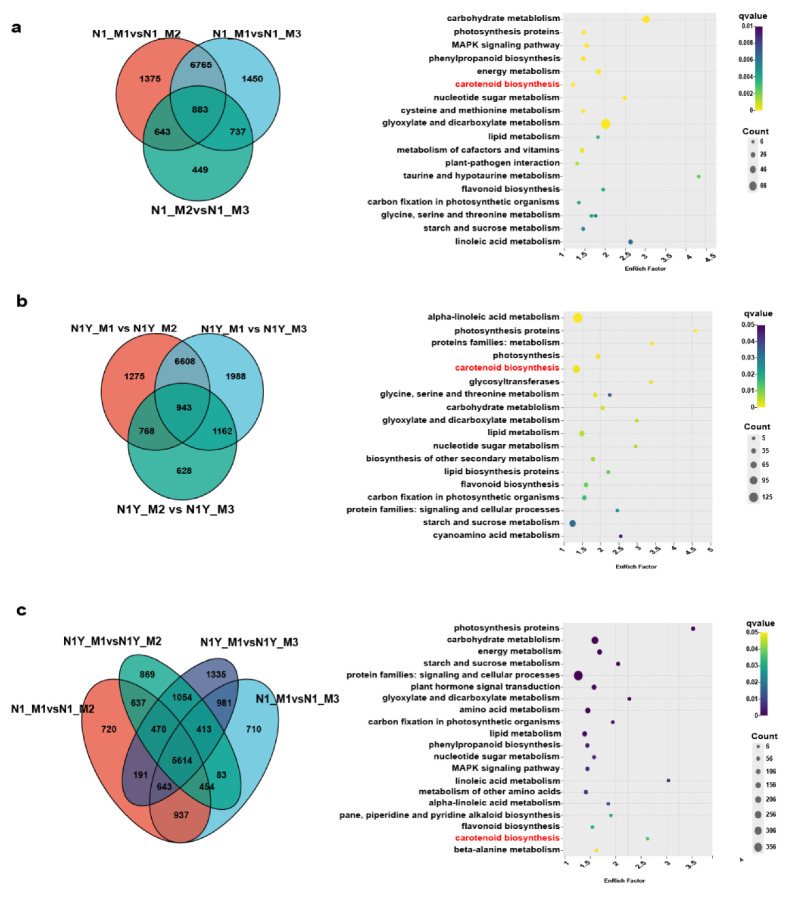
The Venn diagram and KEGG enrichment diagram depict the common DEGs among various samples. (**a**) N1-M1 vs. N1-M2, N1-M vs. N1-M3, and N1-M2 vs. N1-M3. (**b**) N1Y-M1 vs. N1Y-M2, N1Y-M1 vs. N1Y-M3, and N1Y-M2 vs. N1Y-M3. (**c**) N1-M1vs. N1-M2, N1-M1 vs. N1-M3, N1Y-M1 vs. N1Y-M2, and N1Y-M1 vs. N1Y-M3.

**Figure 5 plants-12-02791-f005:**
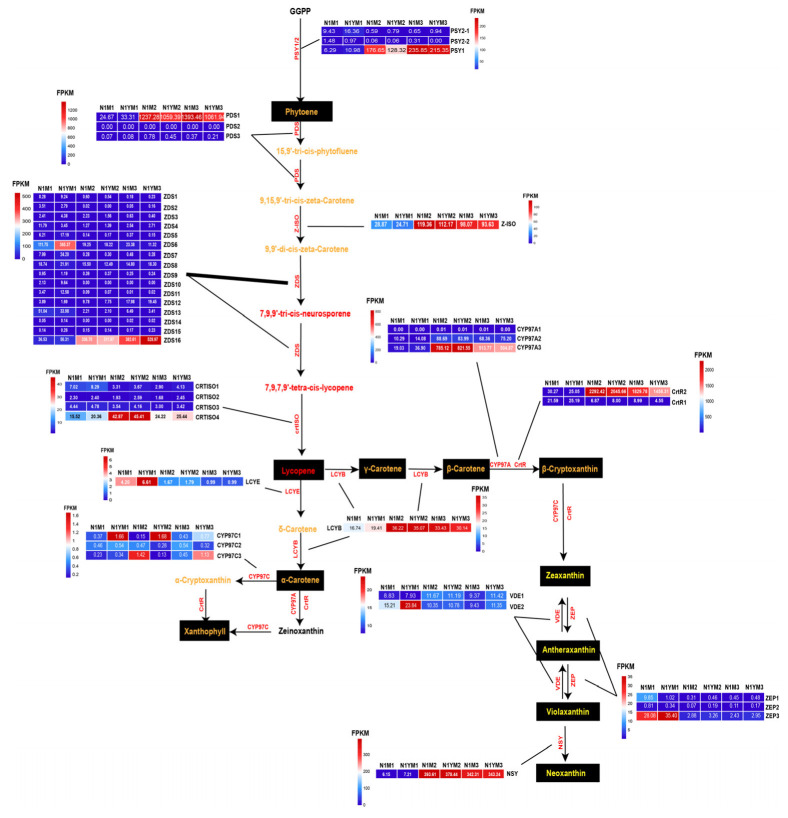
The changes in expression of genes of the carotenoid biosynthetic pathway during ripening of goji fruit. The black background indicates that the substance can be detected in N1 and N1Y fruits. The font color represents the substance color.

**Figure 6 plants-12-02791-f006:**
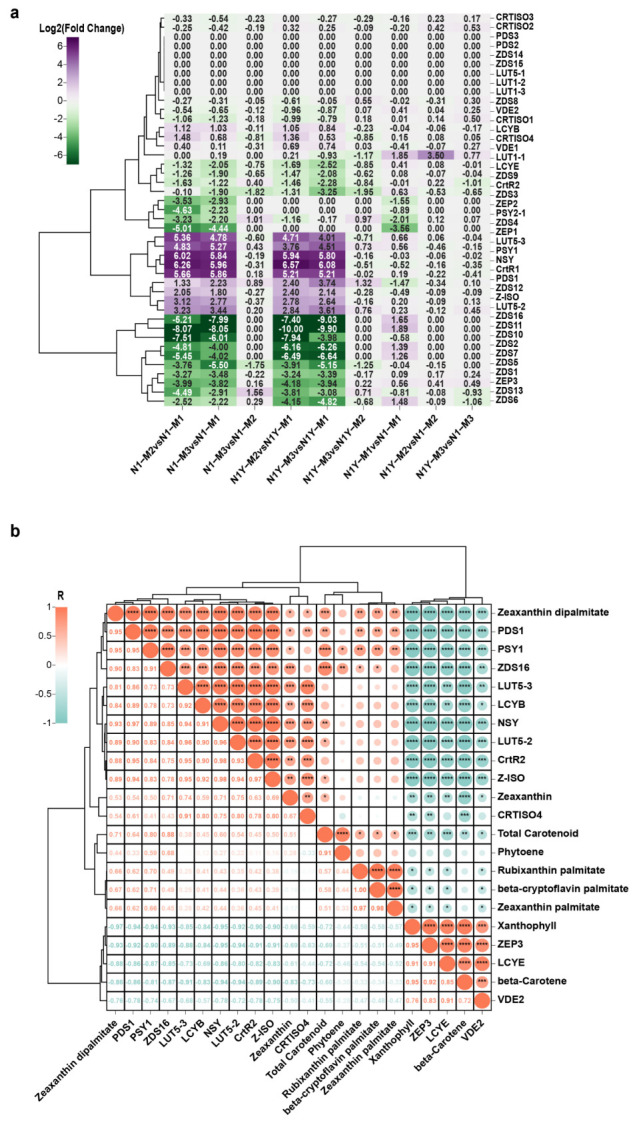
The heatmap of the differences in gene expression in different samples (**a**). The heatmap of the correlation between genes and total carotenoid content (**b**). * Correlation is significant at the 0.05 level. ** Correlation is significant at the 0.01 level. *** Correlation is significant at the 0.001 level. **** Correlation is significant at the 0.0001 level.

**Figure 7 plants-12-02791-f007:**
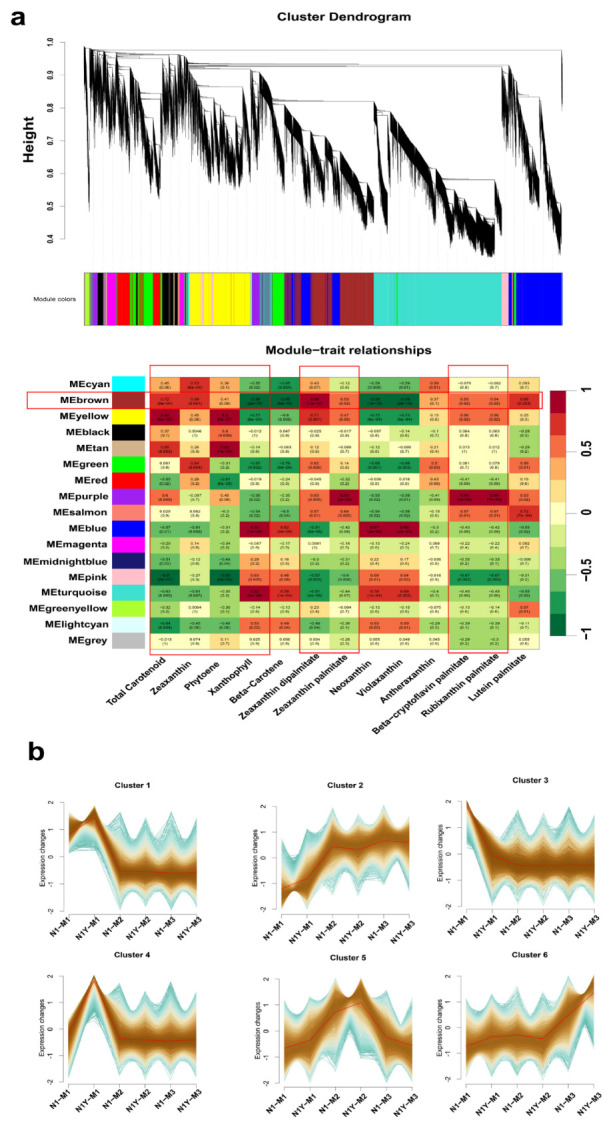
Weighted gene co-expression network analysis (WGCNA) modules of carotenoid biosynthesis in ripening goji fruit (**a**). Gene expression trends analysis of N1 and N1Y fruits during ripening (**b**).

**Figure 8 plants-12-02791-f008:**
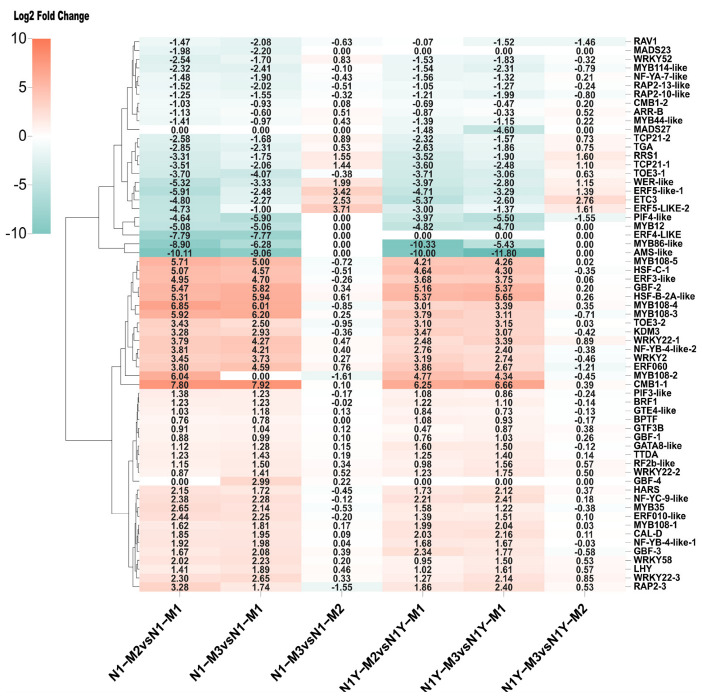
The heatmap of transcription factors expression differences during goji fruit repining.

**Figure 9 plants-12-02791-f009:**
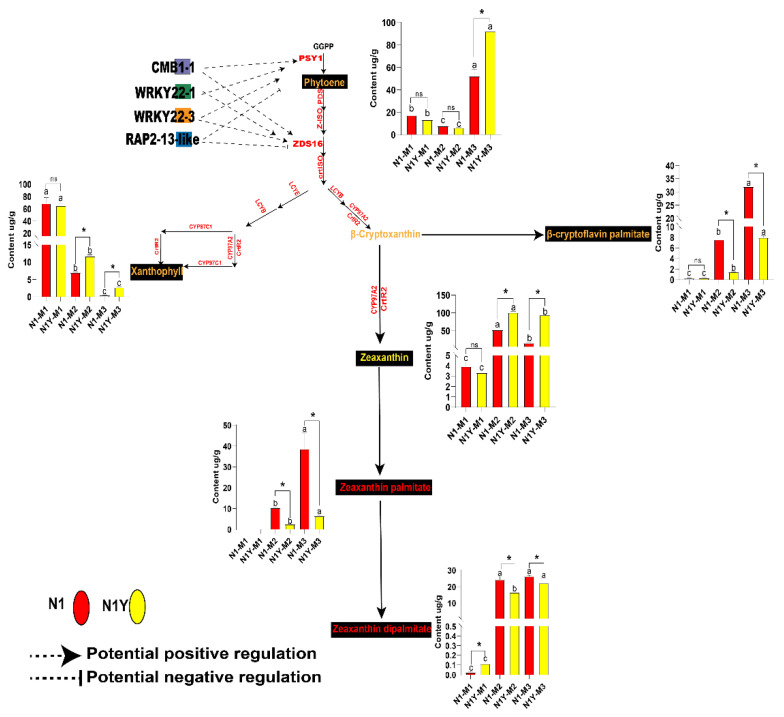
Patterns of transcription factor regulation of carotenoid biosynthesis in goji fruit. The black background indicates that the substance can be detected in N1 and N1Y fruits. The font color represents the substance color. Small letters indicate that N1 or N1Y have significant differences between M1, M2, and M3 (*p* < 0.05), as analyzed by Duncan’s multiple tests; * means a significant difference between N1 and N1Y at M1, M2, and M3 periods (*p* < 0.05), as analyzed by Duncan’s multiple tests; NS means no significant difference.

## Data Availability

The raw RNA-seq data are freely available from the National Center for Biotechnology In-formation (accession: PRJNA946044).

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
