# Peer review of "Transcriptomics and Metabolomics Reveal the Critical Genes of Carotenoid Biosynthesis and Color Formation of Goji (*Lycium barbarum* L.) Fruit Ripening"

_plants, 2023, doi:10.3390/plants12152791_

Round 1

Author Response

Response to Reviewer 1 Comments

Ref: (Plants) Manuscript ID: plants-2497697; MDPI

‘‘Multi-omics reveals the critical genes of carotenoid biosynthesis and color formation of goji (Lycium barbarum L.) fruit ripening’’

Studies seems intriguing as authors have successfully identified 27 carotenoids in Goji fruit and examined their correlation and behavior during various time intervals of maturity (maturity stages). This research presents fascinating findings and offers valuable insights into the synthesis of carotenoids, providing informative knowledge. Overall, the manuscript is written approperiately. The authors have presented all parts in great detail. However, the specific points should be considered before accepting publication.

Point 1: Abstract

Upon reviewing the manuscript, it has been observed that the authors have conducted several analyses. However, it is apparent that the abstract falls short of adequately summarizing the extensive results obtained. Therefore, it is advised that the abstract should be expanded to encompass the entirety of the information, ensuring comprehensive coverage of the subject.

Response 1: Thanks. We have done it per your suggestion. You can find our revision in line 12-24.

Point 2: Introduction

It is well described but may be improved by adding more about the specific carotenoids analyzed in the study and their specific functional significance.

Response 2: Thanks for your suggestions. Because the present study is concerned only with the revelation of the synthesis of carotenoids in goji fruits, and the function of related carotenoids or a specific carotenoid was not studied. Therefore, only the introduction briefly describes the carotenoids on health effects in line 30-31.

Point 3: Material and method are well described.

Response 3: Thanks for your comments.

Point 4: Authors have described all results appropriately and the text is reasonably presented in a scientific way. However, there is need to improve figure quality (Figure 1-5) as the text within them is currently not very readable unfortunately, therefore it is suggested to either split the figures or enhance the text size to ensure better readability.

Response 4: Thanks. We have done it per your suggestion. We split Figure 1-5 into Figures 1- 9. Because word compresses the quality of pictures, we have additionally uploaded the original picture, which we hope you can see it.

Point 5: Line 283-295: Check font size according to journal template

Response 5: Thanks. We have done it per your suggestion.

Point 6: In order to enhance the English grammar in the manuscript, it is desirable to minimize the use of the word "we," which appears repeatedly in the manuscript.

Response 6: Thanks. We have done it per your suggestion.

Reviewer 2 Report

This paper uses a multi-omics approach to conduct an investigation of carotenoid biosynthesis in two different varieties of goji berries, where authors look at carotenoid profiles at different stages of plant development and link profile changes with corresponding genes and transcription factors.

This paper is clearly written, analytical techniques that were chosen are current and sound, results are clearly presented and justified. Data were well explained and presented in variety of graphical ways with statists, and results are discussed and linked with existing research in carotenoid biosynthesis in different species. Below are several areas that could be improved on:

1.       Formatting – several fonts/font sizes are used throughout the text. Some undefined acronyms are present (e.g. BHT), and naming should be consistent – e.g. goji vs wolfberry. Some repletion, portions of introduction are repeated in discussion again. Some repeated sentences -lines 49-51

2.       Profiling of carotenoids is not clearly presented. Reference used in the paper identified 18 carotenoids, the authors quantified 27. Was this profiling based on another paper, or was any additional carotenoid identification work done prior to selecting these 27 standards? Carotenoid standards should be included in method and materials, the raw data for quantification including calibration curves should also be available in SI.

3.       Method – were samples prepared with replicates?

4.       Figures - quality of the figures needs to be improved, either resolution or sizing. Figures should not be grouped the way they are presented, they need to be prioritized or moved to SI. Captions and legends should also be reviewed, some figures lack legends, some images have elements in different colors, where color significance is not explained.

5.       Fig 1 contains 4 sub-figures (a-d), where a, c, and d further contain multiples pictures/graphs. It is impossible to read Fig 1c, and Fig 1d contains 27 individual graphs. Fig 1(b) is trivial as total carotenoid content is expected to increase with maturity; values can easy be reported as text and the graph can be moved to SI. For Fig1(d) several representative carotenoids should be selected, whereas graphs for others should be moved to SI with higher quality.

6.       Fig3 (b) why are some groups in red font? Fig3 (c) – is this figure complete? Legend is not defined, figure itself only includes gene correlation and total carotenoids, which texts discusses gene vs carotenoid correlation. Do more stars correspond to a higher significance level?

7.       Fig 5 –concentration graphs for 5c are useful, but were already included in Fig 1

Author Response

Response to Reviewer 2 Comments

This paper uses a multi-omics approach to conduct an investigation of carotenoid biosynthesis in two different varieties of goji berries, where authors look at carotenoid profiles at different stages of plant development and link profile changes with corresponding genes and transcription factors.This paper is clearly written, analytical techniques that were chosen are current and sound, results are clearly presented and justified. Data were well explained and presented in variety of graphical ways with statists, and results are discussed and linked with existing research in carotenoid biosynthesis in different species. Below are several areas that could be improved on:

Point 1: Formatting – several fonts/font sizes are used throughout the text. Some undefined acronyms are present (e.g. BHT), and naming should be consistent – e.g. goji vs wolfberry. Some repletion, portions of introduction are repeated in discussion again. Some repeated sentences -lines 49-51

Response 1: Thanks. We have done it per your suggestion.

Point 2: Profiling of carotenoids is not clearly presented. Reference used in the paper identified 18 carotenoids, the authors quantified 27. Was this profiling based on another paper, or was any additional carotenoid identification work done prior to selecting these 27 standards? Carotenoid standards should be included in method and materials, the raw data for quantification including calibration curves should also be available in SI.

Response 2: Thanks. We have done it per your suggestion. Determination of carotenoid content was performed by Metware Biotechnology Co., Ltd. (Wuhan, China). Metabolic database of Metware contains 68 carotenoids, but only 27 were detected in goji, so we only show what can be detected in goji fruit. The method for detecting carotenoids in the reference (Liu et al. 2020) we referenced is also the Metware 's method, but because of the difference in species, they only detected 12 in their samples. We have added carotenoid standards and the raw data for quantification including calibration curves in Figure S1.

Reference:

Liu, Y.; Lv, J.; Liu, Z.; Wang, J.; Yang, B.; Chen, W.; Ou, L.; Dai, X.; Zhang, Z.; Zou, X., Integrative analysis of metabolome and transcriptome reveals the mechanism of color formation in pepper fruit (Capsicum annuum L.). Food Chem 2020, 306, 125629.

Point 3: Method – were samples prepared with replicates?

Response 3: All samples had three replicates. Because we also tested other substances, only the sample for carotenoids determination was repeated twice because of insufficient sample. However, according to the QC, the data can explain the problem we have described.

Point 4: Figures - quality of the figures needs to be improved, either resolution or sizing. Figures should not be grouped the way they are presented, they need to be prioritized or moved to SI. Captions and legends should also be reviewed, some figures lack legends, some images have elements in different colors, where color significance is not explained.

Response 4: Thanks. We have done it per your suggestion. We split Figure 1-5 into Figures 1-9. Because word compresses the quality of pictures, we have additionally uploaded the original picture, which we hope you can see it.

Point 5: Fig 1 contains 4 sub-figures (a-d), where a, c, and d further contain multiples pictures/graphs. It is impossible to read Fig 1c, and Fig 1d contains 27 individual graphs. Fig 1(b) is trivial as total carotenoid content is expected to increase with maturity; values can easy be reported as text and the graph can be moved to SI. For Fig1(d) several representative carotenoids should be selected, whereas graphs for others should be moved to SI with higher quality.

Response 5: Thanks for your comments. We split Figure 1 into Figures 1-2. Figure 2 contains 27 individual graphs. We thought that displaying all the detectable carotenoids in goji in a single figure would make it easier for other scholars to read.

Figure 2. The change in individual carotenoid content in N1 and N1Y fruits at different ripening stages. Small letters indicate that N1 or N1Y have significant differences between M1, M2, and M3 (p < 0.05) as analyzed by Duncan’s multiple tests; * means a significant difference between N1 and N1Y at M1, M2, and M3 (p < 0.05), as analyzed by Duncan’s multiple tests; NS means no significant difference. The green, orange, and red asterisks represent the primary carotenoid of N1 and N1Y fruits at the M1, M2, and M3 periods, respectively.

Point 6:  Fig3 (b) why are some groups in red font? Fig3 (c) – is this figure complete? Legend is not defined, figure itself only includes gene correlation and total carotenoids, which texts discusses gene vs carotenoid correlation. Do more stars correspond to a higher significance level?

Response 6: Thanks for your comments. We have redone Figure 3. Here, Figure 3 was split to figure 5 and figure 6. Figure 3(b) Some groups are in red font to make it easier for the reader to notice that we focused on genes that differed in N1 and N1Y during the M2 and M3. We apologize for our inaccurate figure. Figure also contains gene correlation and carotenoids. More stars correspond to a higher significance level

Figure 6. A heatmap of the differences in gene expression in different samples (a). A heatmap of the correlation between genes and total carotenoid content (b). * Correlation is significant at the 0.05 level. ** Correlation is significant at the 0.01 level. *** Correlation is significant at the 0.001 level. **** Correlation is significant at the 0.0001 level.

Point 7: Fig 5 –concentration graphs for 5c are useful, but were already included in Fig 1

Response 7: Thanks for your comments. Actually, figure 5 not only shows the patterns of TF regulation of carotenoid biosynthesis in goji fruit, but also shows the major difference in carotenoids between N1 and N1Y at M1, M2, and M3.

Reviewer 3 Report

“Multi-omics reveals the critical genes of carotenoid biosynthesis and color formation of goji (Lycium barbarum L.) fruit ripening” co-authored by Wei etc. compared the carotenoid content and related gene expression transcriptomic profile between two different colored goli varieties. Multi work had done, but there is still a need for great improvement in writing.

1. Author wrote “multi-omics” in title, but only transcriptomic and target the metabolome were used in study, right? so I suggest just write transcriptomic and metabolome.

2. All Figures can not see clearly. Most of them didn’t have or did not see the the horizontal and vertical axes at all.

3. In Figure 1, The graphs should be arranged from highest to lowest content

4. Line 165-168: what is the color of zeaxanthin palmitate and zeaxanthin dipalmitate ?

5. There are some incorrect font size in paragraph 3.3

6. Line 369, 381, 416, 453 Incorrect citation format

7. which genes are the critical gene caused color difference? Why didn’t further investigated expression of gene related zeaxanthin, zeaxanthin palmitate and zeaxanthin dipalmitate biosynthesis during whole fruit ripening, such as ZEP, CYP97A2.

8. Line 445 Arabidopsis italic

Author Response

Response to Reviewer 3 Comments

“Multi-omics reveals the critical genes of carotenoid biosynthesis and color formation of goji (Lycium barbarum L.) fruit ripening” co-authored by Wei etc. compared the carotenoid content and related gene expression transcriptomic profile between two different colored goli varieties. Multi work had done, but there is still a need for great improvement in writing.

Point 1: Author wrote “multi-omics” in title, but only transcriptomic and target the metabolome were used in study, right? so I suggest just write transcriptomic and metabolome.

Response 1: Thanks. We have done it per your suggestion.

Point 2: All Figures can not see clearly. Most of them didn’t have or did not see the the horizontal and vertical axes at all.

Response 2: Thanks for your comments. We split Figure 1-5 into Figures 1-9. Because word compresses the quality of figures, we have additionally uploaded the original figure, which we hope you can see it.

Point 3: In Figure 1, The graphs should be arranged from highest to lowest content

Response 3: Thanks. We have done it per your suggestion. And we split figure 1 to figure1-2.

Point 4: Line 165-168: what is the color of zeaxanthin palmitate and zeaxanthin dipalmitate?

Response 4: Thanks for your comments. The color of zeaxanthin palmitate and zeaxanthin dipalmitate was red.

Point 5: There are some incorrect font size in paragraph 3.3

Response 5: Thanks. We have done it per your suggestion.

Point 6: Line 369, 381, 416, 453 Incorrect citation format

Response 6: Thanks. We have done it per your suggestion.

Point 7: which genes are the critical gene caused color difference? Why didn’t further investigated expression of gene related zeaxanthin, zeaxanthin palmitate and zeaxanthin dipalmitate biosynthesis during whole fruit ripening, such as ZEP, CYP97A2.

Response 7: Thanks for your comments. Although, we want to found the critical gene caused color difference between N1 and N1Y. Due to how zeaxanthin converted to zeaxanthin palmitate and zeaxanthin dipalmitate is still unclear, and the key gene was no report. So, we did not find the critical gene cause color difference between N1 and N1Y according to our data. We only found that differences in the main carotenoids can explain the color variation between N1 and N1Y fruit at the M2 and M3 stages. Much of the zeaxanthin in the N1 fruit was converted to zeaxanthin palmitate and zeaxanthin dipalmitate during the M2 and M3 periods, while the N1Y fruit continued to accumulate zeaxanthin. As a result, from M2 to M3, the N1 fruit gradually turns red, whereas the N1Y fruit turns yellow.

Although, we found some critical gene (PSY1 and ZDS16) and TF are involved in carotenoid biosynthesis of N1 and N1Y fruit. But those critical gene have no difference between N1 and N1Y at the difference periods.

Point 8: Line 445 Arabidopsis italic

Response 8: Thanks. We have done it per your suggestion.

Reviewer 4 Report

The carotenoids in Lycium barbarum have good health effects, but the underlying mechanism of carotenoid synthesis and color formation during fruit ripening is still unclear. In this manuscript, the authors used transcriptomics and carotenoid analysis to investigate the differences in carotenoid biosynthesis and color formation of red fruit N1 and yellow fruit N1Y at three stages of maturation. Finally, PSY1 and ZDS16 play an important role in carotenoid biosynthesis during the ripening of Lycium berries, revealing the molecular network of carotenoid synthesis and explaining the differences in carotenoid accumulation in different colors of Lycium berries. However, I think there are still the following problems with this manuscript:

1.There is a problem with the format of lines 283-295. Please modify it carefully;

2.There are error characters in 600-601lines, please carefully modify;

3. Whether PSY1 and ZDS16 are represented in italics, please check the full text carefully;

4.N1fruitsin Line 156 should be separated by Spaces;

5. The unit ug/g of flavonoid content in Fig1b should be bracketed;

6. at M1, M2, and M3 in the legend should be changed to at the M1, M2, and M3 periods;

7.191 A period should be added at the end of the line;

8. The text size, character spacing is inconsistent, and the picture clarity is low, it is recommended that the author carefully check;

9. Some of the text is expressed in the past tense, some in the present tense, such as lines 320 and 321, please check the full text carefully;

10. Whether a and b in the legend need bold, suggest the author to modify the format;

11. “In the M3 period” in line 377 should be changed to “ At the M3 period”.

Authors are advised to check the following format carefully.

Author Response

Response to Reviewer 4 Comments

The carotenoids in Lycium barbarum have good health effects, but the underlying mechanism of carotenoid synthesis and color formation during fruit ripening is still unclear. In this manuscript, the authors used transcriptomics and carotenoid analysis to investigate the differences in carotenoid biosynthesis and color formation of red fruit N1 and yellow fruit N1Y at three stages of maturation. Finally, PSY1 and ZDS16 play an important role in carotenoid biosynthesis during the ripening of Lycium berries, revealing the molecular network of carotenoid synthesis and explaining the differences in carotenoid accumulation in different colors of Lycium berries. However, I think there are still the following problems with this manuscript:

Point 1: There is a problem with the format of lines 283-295. Please modify it carefully;

Response 1: Thanks for your comments. We have done it per your suggestion.

Point 2: There are error characters in 600-601lines, please carefully modify;

Response 2: Thanks. We have done it per your suggestion.

Point 3: Whether PSY1 and ZDS16 are represented in italics, please check the full text

carefully;

Response 3: Thanks for your comments. We have done it per your suggestion.

Point 4: “N1fruits”in Line 156 should be separated by Spaces;

Response 4: Thanks for your comments. We have done it per your suggestion.

Point 5: The unit ug/g of flavonoid content in Fig1b should be bracketed;

Response 5: Thanks. We have done it per your suggestion.

Point 6: at M1, M2, and M3 in the legend should be changed to at the M1, M2, and M3 periods;

Response 6: Thanks for your comments. We have done it per your suggestion.

Point 7: 191 A period should be added at the end of the line;

Response 7: Thanks. We have done it per your suggestion.

Point 8: The text size, character spacing is inconsistent, and the picture clarity is low, it is recommended that the author carefully check;

Response 8: Thanks for your comments. We split Figure 1-5 into Figures 1-9. Because word compresses the quality of figures, we have additionally uploaded the original figure, which we hope you can see it.

Point 9: Some of the text is expressed in the past tense, some in the present tense, such as lines 320 and 321, please check the full text carefully;

Response 9: Thanks. We have done it per your suggestion.

Point 10: Whether a and b in the legend need bold, suggest the author to modify the format;

Response 10: Thanks. We have done it per your suggestion.

Point 11: “In the M3 period” in line 377 should be changed to “ At the M3 period”.

Response 11: Thanks. We have done it per your suggestion.
